# RP1M: A Large-Scale Motion Dataset for Piano Playing with Bimanual Dexterous Robot Hands

**Yi Zhao** [*†1]   **Le Chen** [*2]   **Jan Schneider** [2]   **Quankai Gao** [3]
**Juho Kannala** [1,4]   **Bernhard Schölkopf** [2]   **Joni Pajarinen** [1]   **Dieter Büchler** [2]
[1]Aalto University, Finland   [2]Max Planck Institute for Intelligent Systems, Germany
[3]University of Southern California, USA   [4]University of Oulu, Finland

**Abstract:** It has been a long-standing research goal to endow robot hands with human-level dexterity. Bimanual robot piano playing constitutes a task that combines challenges from dynamic tasks, such as generating fast while precise motions, with slower but contact-rich manipulation problems. Although reinforcement learning-based approaches have shown promising results in single-task performance, these methods struggle in a multi-song setting. Our work aims to close this gap and, thereby, enable imitation learning approaches for robot piano playing at scale. To this end, we introduce the *Robot Piano 1 Million* (RP1M) dataset, containing bimanual robot piano playing motion data of more than one million trajectories. We formulate finger placements as an optimal transport problem, thus, enabling automatic annotation of vast amounts of unlabeled songs. Benchmarking existing imitation learning approaches shows that such approaches reach promising robot piano playing performance by leveraging RP1M [◇].

**Keywords:** Bimanual dexterous robot hands, dataset for robot piano playing, imitation learning, robot learning at scale

## 1   Introduction

Empowering robots with human-level dexterity is notoriously challenging. Current robotics research on hand and arm motions focuses on manipulation and dynamic athletic tasks. Manipulation, such as grasping or reorienting [1], requires continuous application of acceptable forces at moderate speeds to various objects with distinct shapes and weight distributions. Environmental changes, like humidity or temperature, alter the already complex contact dynamics, which adds to the complexity of manipulation tasks. Dynamic tasks, like juggling [2] and table tennis [3] involve making and breaking contact, demanding high precision and tolerating less inaccuracy due to rarer contacts. High speeds in these tasks necessitate greater accelerations and introduce a precision-speed tradeoff.

Robot piano playing combines various aspects of dynamic and manipulation tasks: the agent is required to coordinate multiple fingers to precisely press keys for arbitrary songs, which is a high-dimensional and rich control task. At the same time, the finger motions have to be highly dynamic, especially for songs with fast rhythms. Well-practiced pianists can play arbitrary songs, but this level of generalization is extremely challenging for robots. In this work, we build the foundation to develop methods capable of achieving human-level bi-manual dexterity at the intersection of manipulation and dynamic tasks, while reaching such generalization capabilities in multi-task environments.

While reinforcement learning (RL) is a promising direction, traditional RL approaches often struggle to achieve excellent performance in multi-task settings [4]. The advent of scalable imitation learning techniques [5] enables representing complex and multi-modal distributions. Such large models

---

[*] Equal contribution. Correspondence to `yi.zhao@aalto.fi`, `le.chen@tuebingen.mpg.de`.

[†] Work done during an internship at Max Planck Institute for Intelligent Systems.

[◇] Project website: `https://rp1m.github.io/`

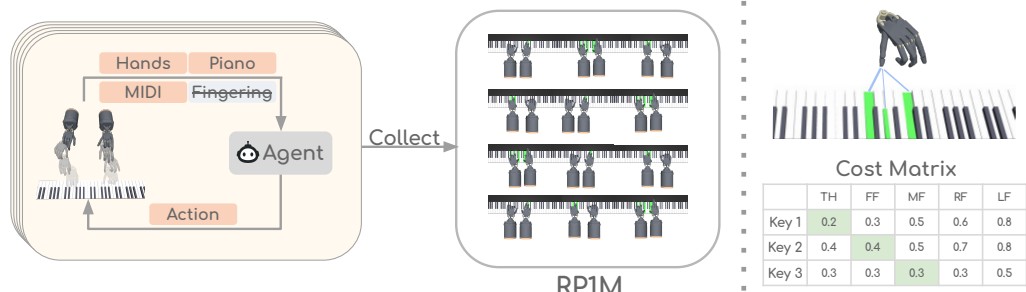

Figure 1: Overview of RP1M. (**Left**) RP1M is a large-scale motion dataset for piano playing with bi-manual dexterous robot hands. The dataset includes ∼*1M* expert trajectories collected by ∼*2k* RL specialist agents. (**Right**) To collect a diverse motion dataset of playing sheet music available on the Internet, we lift the requirement of human-annotated fingering by formulating the finger placement as an optimal transport problem such that the robot hands play piano in an energy-efficient way.

are most effective when trained on massive datasets that combine the state evolution with the corresponding action trajectories. So far, creating large datasets for robot piano play is problematic due to the time-consuming fingering annotations. Fingering annotations map which finger is supposed to press a particular piano key at each time step. With fingering information, the reward is less sparse, making the training significantly more effective. These labels usually require expert human annotators [6], preventing the agent from leveraging the large amounts of unlabeled music pieces on the internet [7]. Besides, human-labeled fingering may be infeasible for robots with morphologies different from human hands, such as different numbers of fingers or distinct hand dimensions.

In this paper, we propose the *Robot Piano 1 Million dataset* (RP1M). This dataset comprises the motion data of high-quality bi-manual robot piano play. In particular, we train RL agents for each of the 2k songs and roll out each policy 500 times with different random seeds. To enable the generation of RP1M, we introduce a method to learn the fingering automatically by formulating finger placement as an optimal transport (OT) problem [8, 9]. Intuitively, the fingers are placed in a way such that the correct keys are pressed while the overall moving distance of the fingers is minimized. Agents trained using our automatic fingering match the performance of agents trained with human-annotated fingering labels. Besides, our method is easy to implement with almost no extra training time. The automatic fingering also allows learning piano playing with different embodiments, such as robots with four fingers only. With RP1M, it is now possible to train and test imitation learning approaches at scale. We benchmark various behavior cloning approaches and find that using RP1M, existing methods perform better in terms of generalization capabilities in multi-song piano play. This work contributes in various ways:

- To facilitate the research on dexterous robot hands, we release *RP1M*, a dataset of piano playing motions that includes more than 2k music pieces with expert trajectories generated by our agents.
- We formulate fingering as an optimal transport problem, enabling the generation of vast amounts of robot piano data with RL, as well as allowing variations in the embodiment.
- Using RP1M, we benchmark various approaches to robot piano playing, whereby existing imitation learning approaches reach promising results in motion synthesis on novel music pieces due to scaling with RP1M.

## 2   Related Work

**Piano Playing with Robots**   Piano playing with robot hands has been investigated for decades. It is a challenging task since bimanual robot hands should precisely press the right keys at the right time, especially considering its high-dimensional action space. Previous methods require specific robot designs [10, 11, 12, 13, 14, 15] or trajectory pre-programming [16, 17]. Recent methods enable piano playing with dexterous hands through planning [18] or RL [19] but are limited to simple music pieces. RoboPianist [4] introduces a benchmark for robot piano playing and demonstrates strong RL performance, but requires human fingering labels and performs worse in multi-task learning.

Table 1: Existing datasets on dexterous or bimanual robotic manipulation.

| Dataset | Task | Dexterous hands | Bimanual | Dynamic tasks | Demonstrations |
|---|---|---|---|---|---|
| DexGraspNet [28] | grasping | ✓ | | | 1.3M |
| RealDex [31] | grasping | ✓ | | | 2.6K |
| UniDexGrasp [30] | grasping | ✓ | | | 1.1M |
| ALOHA [33] | manipulation | | ✓ | | 825 |
| Bi-DexHands [34] | manipulation | ✓ | ✓ | partially | ~20K |
| D4RL [32] (Adroit) | manipulation | ✓ | | | 30K |
| RP1M (ours) | piano | ✓ | ✓ | ✓ | 1M |

Yang et al. [20] improves the policy training performance by considering the bionic constraints of the anthropomorphic robot hands.

Human fingering informs the agent of the correspondence between fingers and pressed keys at each time step. These labels require expert annotators and are, therefore, expensive to acquire in practice. Several approaches learn fingering from human-annotated data with different machine learning methods [6, 21, 22]. Moryossef et al. [23] extract fingering from videos to acquire fingering labels cheaply. Ramoneda et al. [24] proposes to treat piano fingering as a sequential decision-making problem and use RL to calculate fingering but without considering the model of robot hands. Shi et al. [25] automatically acquires fingering via dynamic programming, but the solution is limited to simple tasks. Concurrent work [26] obtains fingering labels from YouTube videos and trains a diffusion policy to play hundreds of songs. In our paper, we do not introduce a separate fingering model, instead, similar to human pianists, fingering is *discovered automatically* while playing the piano, hereby largely expanding the pool of usable data to train a generalist piano-playing agent.

**Datasets for Dexterous Robot Hands**  Most large-scale datasets of dexterous robot hands focus on grasping various objects. To get suitable grasp positions, some methods utilize planners [27, 28, 29], while others use learned grasping policies [30], or track grasping motions of humans and imitate these motions on a robot hand [31]. Compared to the abundance of datasets for grasping, there exist relatively few datasets for object manipulation with dexterous robot hands. The D4RL benchmark [32] provides small sets of expert trajectories for four such tasks, consisting of human demonstrations and rollouts of trained policies. Zhao et al. [33] provides a small object manipulation dataset that utilizes a low-cost bimanual platform with simple parallel grippers. Chen et al. [34] collects offline datasets for two simulated bimanual manipulation tasks with dexterous hands. Furthermore, Fan et al. [35] proposes a large-scale dataset for bimanual hand-object manipulation with human hands rather than robot hands. Table 1 summarizes the characteristics of these existing datasets. To the best of our knowledge, our RP1M dataset is the first large-scale dataset of dynamic, bimanual piano playing with dexterous robot hands.

We further discuss related work on dexterous robot hands and generalist agents in Appendix A.

## 3  Background

**Task Setup**  The simulated piano-playing environment is built upon RoboPianist [4]. It includes a robot piano-playing setup, an RL-based agent for playing piano with simulated robot hands, and a multi-task learner. To avoid confusion, we refer to these components as *RoboPianist*, *RoboPianist-RL*, and *RoboPianist-MT*, respectively. The piano playing environment features a full-size keyboard with 88 keys driven by linear springs, two Shadow robot hands [36], and a pseudo sustain pedal.

Sheet music is represented by Musical Instrument Digital Interface (MIDI) transcription. Each time step in the MIDI file specifies which piano keys to press (active keys). The goal of a piano-playing agent is to press active keys and avoid inactive keys under *space* and *time* constraints. This requires the agent to coordinate its fingers and place them properly in a highly dynamic scenario such that

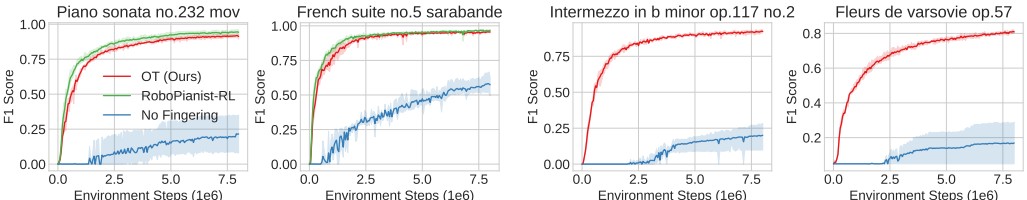

Figure 2: Comparison of the RL performance with our OT fingering, human-annotated fingering, and no fingering. Our method matches the performance of RoboPianist-RL, which is trained with human fingering. We also outperforms the baseline without any fingering information by a large margin. The plots show the mean over 3 random seeds and the shaded areas represent the 95% confidence interval.

target keys, at not only the current time step but also the future time steps, can be pressed accurately and timely. The original RoboPianist uses MIDI files from the PIG dataset [6] which includes *human fingering* information annotated by experts. However, as mentioned earlier, this limits the agent to only play human-labeled music pieces, and the human annotation may not be suitable for robots due to the different morphologies.

The observation includes the state of the two robot hands, fingertip positions, piano sustain state, piano key states, and a goal vector, resulting in an 1144-dimensional observation space. The goal includes 10-step active keys and 10-step target sustain states obtained from the MIDI file, represented by a binary vector. RoboPianst further includes 10-step human-labeled fingering in the observation space but we remove this observation in our method since we do not need human-labeled fingering. For the action space, we remove the DoFs that do not exist in the human hand or are used in most songs, resulting in a 39-dimensional action space, consisting of the joint positions of the robot hands, the positions of forearms, and a sustain pedal. We evaluate the performance of the trained agent with an average F1 score calculated by $F_1 = 2 \cdot \frac{\text{precision} \cdot \text{recall}}{\text{precision} + \text{recall}}$. For piano playing, recall and precision measure the agent's performance on pressing the active keys and avoiding inactive keys respectively [4].

**Playing Piano with RL**   We use RL to train specialist agents per song to control the bimanual dexterous robot hands to play the piano. We frame the piano playing task as a finite Markov Decision Process (MDP). At time step $t$, the agent $\pi_\theta(a_t|s_t)$, parameterized by $\theta$, receives state $s_t$ and takes action $a_t$ to interact with the environment and receives new state $s_{t+1}$ and reward $r_t$. The state and action spaces are described above and the reward $r_t$ gives an immediate evaluation of the agent's behavior. We will introduce reward terms used for training in Section 4.1. The agent's goal is to maximize the expected cumulative rewards over an episode of length $H$, defined as $\mathcal{J} = \mathbb{E}_{\pi_\theta} \left[ \sum_{t=0}^{H} \gamma^t r_t(s_t, a_t) \right]$, where $\gamma$ is a discount factor ranging from 0 to 1.

## 4  Large-Scale Motion Dataset Collection

In this section, we describe our RP1M dataset in detail. We first introduce how to train a specialist piano-playing agent without human fingering labels. Removing the requirement of human fingering labels allows the agent to play any sheet music available on the Internet (under copyright license). We then analyze the performance of our specialist RL agent as well as the learned fingering. Lastly, we introduce our collected large-scale motion dataset, RP1M, which includes ~1M expert trajectories for robot piano playing, covering ~2k pieces of music.

### 4.1  Piano Playing without Human Fingering Labels

To mitigate the hard exploration problem posed by the sparse rewards, RoboPianist-RL adds dense reward signals by using human fingering labels. Fingering informs the agent of the "ground-truth" fingertip positions, and the agent minimizes the Euclidean distance between the current fingertip positions and the "ground-truth" positions. We now discuss our OT-based method to lift the requirement of human fingering.

Although fingering is highly personalized, generally speaking, it helps pianists to press keys timely and efficiently. Motivated by this, apart from maximizing the key pressing rewards, we also aim to minimize the moving distances of fingers. Specifically, at time step $t$, for the $i$-th key $k^i$ to press, we use the $j$-th finger $f^j$ to press this key such that the overall moving cost is minimized. We define the minimized cumulative moving distance between fingers and target keys as $d_t^{OT} \in \mathbb{R}^+$, given by

$$
\begin{aligned}
d_t^{OT} = \min_{w_t} \sum_{(i,j) \in K_t \times F} w_t(k^i, f^j) \cdot c_t(k^i, f^j), \ \ \text{s.t.,} \ \ i) \sum_{j \in F} w_t(k^i, f^j) = 1, \ \ \text{for} \ \ i \in K_t, \\
ii) \sum_{i \in K_t} w_t(k^i, f^j) \leq 1, \ \ \text{for} \ \ j \in F, \ \ iii) \ w_t(k^i, f^j) \in \{0, 1\}, \ \ \text{for} \ \ (i,j) \in K_t \times F.
\end{aligned}
\tag{1}
$$

$K_t$ represents the set of keys to press at time step $t$ and $F$ represents the fingers of the robot hands. $c_t(k^i, f^j)$ represents the cost of moving finger $f^j$ to piano key $k^i$ at time step $t$ calculated by their Euclidean distance. $w_t(k^i, f^j)$ is a boolean weight. In our case, it enforces that each key in $K_t$ will be pressed by only *one* finger in $F$, and each finger presses *at most* one key. The constrained optimization problem in Eq. (1) is an optimal transport problem. Intuitively, it tries to find the best "transport" strategy such that the overall cost of moving (a subset of) fingers $F$ to keys $K_t$ is minimized. We solve this optimization problem with a modified Jonker-Volgenant algorithm [37] from SciPy [38] and use the optimal combinations $(i^*, j^*)$ as the fingering for the agent. The fingering is calculated on the fly based on the hands' state, so during the RL training, the fingering adjusts according to the robot hands' state.

We define a reward $r_t^{OT}$ to encourage the agent to move the fingers close to the keys $K_t$. which is defined as:

$$
r_t^{OT} = \begin{cases} \exp(c \cdot (d_t^{OT} - 0.01)^2) & \text{if } d_t^{OT} \geq 0.01, \\ 1.0 & \text{if } d_t^{OT} < 0.01. \end{cases}
\tag{2}
$$

$c$ is a constant scale value as used in Tassa et al. [39] and $d_t^{OT}$ is the distance between fingers and target keys obtained by solving Eq. (1). $r_t^{OT}$ increases exponentially as $d_t^{OT}$ decreases and is set as 1 once $d_t^{OT}$ is smaller than a pre-defined threshold (0.01). The overall reward function is defined as:

$$
r_t = r_t^{OT} + r_t^{Press} + r_t^{Sustain} + \alpha_1 \cdot r_t^{Collision} + \alpha_2 \cdot r_t^{Energy}
\tag{3}
$$

$r^{Press}$ and $r_t^{Sustain}$ represent the reward for correctly pressing the target keys and the sustain pedal. $r_t^{Collision}$ encourages the agent to avoid collision between forearms and $r_t^{Energy}$ prefers energy-saving behaviors. $\alpha_1$ and $\alpha_2$ are coefficient terms, and $\alpha_1 = 0.5$ and $\alpha_2 = 5 \cdot 10^{-3}$ are adopted. Our method is compatible with any RL methods, and we use DroQ [40] in our paper.

### 4.2 Analysis of Specialist RL Agents

The performance of the specialist RL agents decides the quality of our dataset. In this section, we investigate the performance of our specialist RL agents. We are interested in i) how the proposed OT-based finger placement helps learning, ii) how the fingering discovered by the agent itself compares to human fingering labels, and iii) how our method transfers to other embodiments.

**Results** In Fig. 2, we compare our method with RoboPianist-RL both with and without human fingering. We use the same DroQ algorithm with the same hyperparameters for all experiments. RoboPianist-RL includes human fingering in its inputs, and the fingering information is also used in the reward function to force the agent to follow this fingering. Our method, marked as *OT*, removes the fingering from the observation space and uses OT-based finger placement to guide the agent to discover its own fingering. We also include a baseline, called *No Fingering*, that removes the fingering entirely. The first two columns of Fig. 2 show that our method without human-annotated fingering matches RoboPianst-RL's performance on two different songs. Our method outperforms the baseline without human fingering by a large margin, showing that the proposed OT-based finger placement boosts the agent learning. The proposed method works well even on challenging songs. We test our method on *Flight of the Bumblebee* and achieve 0.79 F1 score after 3M training steps. To

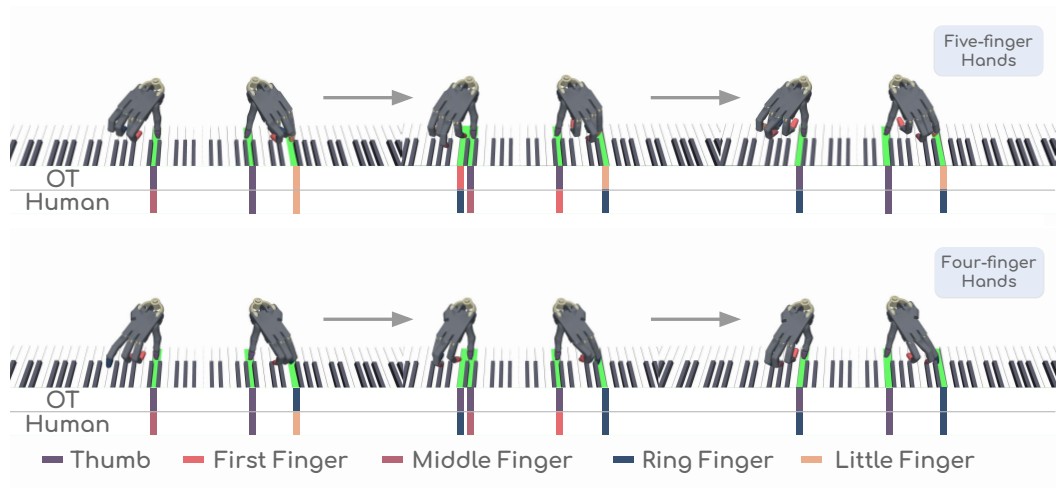

Figure 3: Comparison of fingering discovered by the agent itself and human annotations.

the best of our knowledge, we are the first to play the challenging song Flight of the Bumblebee with general-purpose bimanual dexterous robot hands.

**Analysis of the Learned Fingering**   We now compare the fingering discovered by the agent itself and the human annotations. In Fig. 3, we visualize the sample trajectory of playing *French Suite No.5 Sarabande* and the corresponding fingering. We found that although the agent achieves strong performance for this song (the second plot in Fig. 2), our agent discovers different fingering compared to humans. For example, for the right hand, humans mainly use the middle and ring fingers, while our agent uses the thumb and first finger. Furthermore, in some cases, human annotations are not suitable for the robot hand due to different morphologies. For example, in the second time step of Fig. 3, the human uses the first finger and ring finger. However, due to the mechanical limitation of the robot hand, it can not press keys that far apart with these two fingers, thus mimicking human fingering will miss one key. Instead, our agent discovered to use the thumb and little finger, which satisfies the hardware limitation and accurately presses the target keys.

**Cross Emboidments**   Labs usually have different robot platforms, thus having a method that works for different embodiments is highly desirable. We test our method on a different embodiment. To simplify the experiment, we disable the little finger of the Shadow robot hand and obtain a four-finger robot hand, which has a similar morphology to Allegro [41] and LEAP Hand [42]. We evaluate the modified robot hand on the song French Suite No.5 Sarabande (first 550 time steps), where our method achieves a 0.95 F1 score, similar to the 0.96 achieved with the original robot hands. In the bottom row of Fig. 3, we visualize the learned fingering with four-finger hands. The agent discovers different fingering compared to humans and the original hands but still accurately presses active keys, meaning our method is compatible with different embodiments.

### 4.3   RP1M Dataset

To facilitate the research on dexterous robot hands, we collect and release a large-scale motion dataset for piano playing. Our dataset includes ∼1M expert trajectories covering ∼2k musical pieces. For each musical piece, we train an individual DroQ agent with the method introduced in Section 4.1 for 8 million environment steps and collect 500 expert trajectories with the trained agent. We chunk each sheet music every 550 time steps, corresponding to 27.5 seconds, so that each run has the same episode length. The sheet music used for training is from the PIG dataset [6] and a subset (1788 pieces) of the GiantMIDI-Piano dataset [7].

In Fig. 4, we show the statistics of our collected motion dataset. The top plot shows the histogram of the pressed keys. We found that keys close to the center are more frequently pressed than keys at the corner. Also, white keys, taking 65.7%, are more likely to be pressed than black keys. In the bottom left plot, we show the distribution of the number of active keys over all time steps. It roughly follows

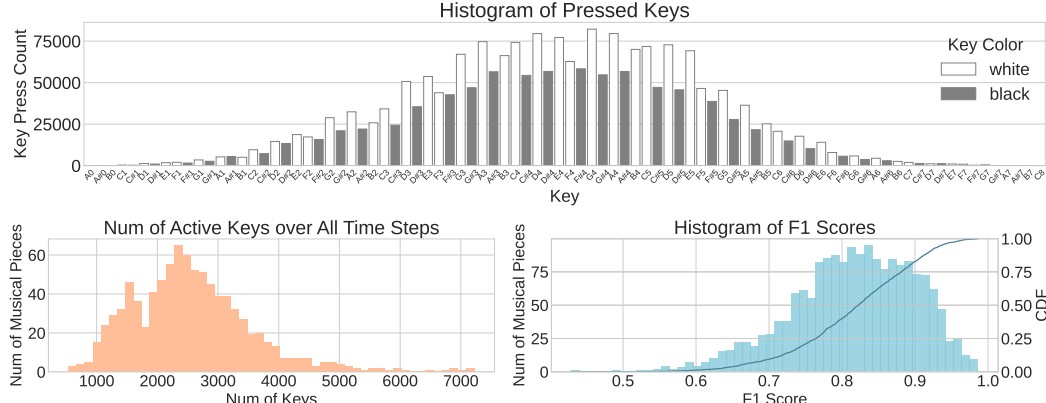

Figure 4: Statistics of our RP1M dataset. (**Top**) Histogram of pressed keys in our RP1M dataset. (**Bottom Left**) Distribution of the number of active keys over all time steps. (**Bottom Right**) Distribution of F1 scores in our dataset.

a Gaussian distribution, and 90.70% musical pieces in our dataset include 1000-4000 active keys. We also include the distribution of F1 scores of trained agents used for collecting data. We found most agents (79.00%) achieve F1 scores larger than 0.75, and 99.89% of the agents' F1 scores are larger than 0.5. The distribution of F1 scores reflects the quality of the collected dataset. We empirically found agents with F1 score $\geq 0.75$ are capable of playing sheet music reasonably well with only minor errors. Agents with $\leq 0.5$ F1 scores usually have notable errors due to the difficulty of songs or the mechanical limitations of the Shadow robot hand. We also include the F1 scores for each piece in our dataset so users can filter the dataset according to their needs.

## 5 Benchmarking Results

The analysis in the previous section highlighted the diversity of highly dynamic piano-playing motions in the RP1M dataset. In this section, we assess the multi-task imitation learning performance of several widely used methods on our benchmark. To be specific, the objective is to train a single multi-task policy capable of playing various music pieces on the piano. We train the policy on a portion of the RP1M dataset and evaluate its in-distribution performance (F1 scores on songs included in the training data) and its generalization ability (F1 scores on songs not present in the training data).

**Baselines** We evaluated Behavior Cloning (BC) [43], Behavior Transformer (BeT) [44], Diffusion Policy [5] with U-Net (DP-U) [45] and with Transformer (DP-T) [46]. BC directly learns a policy by using supervised learning on observation-action pairs from expert demonstrations. BeT clusters continuous actions into discrete bins using k-means, allowing it to model high-dimensional, continuous, multimodal action distributions as categorical distributions [44]. Diffusion Policy learns to model the action distribution by inverting a process that gradually adds noise to a sampled action sequence. We evaluated both the CNN-based (U-Net) Diffusion Policy (DP-U) and the Transformer-based Diffusion Policy (DP-T) with DDPM [47]. We use the same code and hyperparameters as Chi et al. [5]. Detailed descriptions of the baselines as well as hyperparameters are given in Appendix C.1.

**Experiment Setup** We train the policies on subsets of the RP1M dataset with different sizes: 12, 25, 50, 100, 150. We then evaluate the trained policies on both i) 12 in-distribution songs: music pieces that overlap with the training sets, and ii) 20 out-of-distribution (OOD) songs: music pieces that do not overlap with the training songs. The selected songs are very challenging and contain diverse motions and long horizons. In the experiment, we report zero-shot evaluation results without fine-tuning. We report the average F1 scores of each group of music pieces for policies trained with each baseline method. We list the selected songs for evaluation in Appendix C.2.

**Discussion** We present the benchmarking performance of multi-task agents in Table 2. For the *in-distribution* evaluation, compared to F1 scores obtained with our RL specialist agents in Fig. 4, we notice a performance gap across all baselines. This gap widens as the data size increases. When

Table 2: Comparison results of multi-task imitation learning.

| # music | In-disribution | | | | | Out-of-distribution | | | | |
|---|---|---|---|---|---|---|---|---|---|---|
| | 12 | 25 | 50 | 100 | 150 | 12 | 25 | 50 | 100 | 150 |
| **BC-MLP** | 0.529 | 0.315 | 0.319 | 0.193 | 0.250 | 0.079 | 0.119 | 0.100 | 0.108 | 0.187 |
| **BeT** | 0.062 | 0.080 | 0.065 | 0.078 | 0.088 | 0.094 | 0.110 | 0.111 | 0.120 | 0.125 |
| **DP-U** | 0.539 | 0.541 | 0.546 | 0.505 | 0.454 | 0.181 | 0.189 | 0.198 | 0.215 | 0.256 |
| **DP-T** | 0.357 | 0.304 | 0.297 | 0.301 | 0.318 | 0.186 | 0.210 | 0.230 | 0.291 | 0.316 |

trained on a smaller dataset with 12 training songs, DP-U performs comparably to BC-MLP and slightly outperforms DP-T, while BeT experiences a significant performance drop. This decline may be attributed to hyperparameter choices, such as the number of action bins. Although we used the same number of action bins as the official implementation, the complexity of our tasks suggests that this configuration may be inadequate, and increasing the number of bins could improve performance.

As the dataset size increases, we observe that Diffusion Policy outperforms the other baselines. DP-U and DP-T show performance drops of 15.77% and 10.92%, respectively, while BC-MLP suffers a more significant decline of 52.74%. Similar performance gaps have been noted in previous work [4] and concurrent research [26] suggests a hierarchical policy structure, although it still lags behind RL specialists. This highlights the need for future research to address the performance gap between RL specialists and multi-task agents.

In the *zero-shot out-of-distribution* evaluation, we find that performance improves for all evaluated baselines as the training data size increases. Specifically, the F1 scores for DP-U and DP-T rise from 0.181 to 0.256 and from 0.186 to 0.316, respectively, when the number of training songs is increased from 12 to 150. This suggests that larger datasets enhance the generalization capabilities of multi-task agents. We hope that releasing our large-scale RP1M dataset will contribute to the development of robust generalist piano-playing agents within the research community.

## 6 Limitations & Conclusion

**Limitations** Our paper has limitations in several aspects. Firstly, although our method lifts the requirement of human-annotated fingering, enabling RL training on diverse songs, our method still fails to achieve strong performance on challenging songs due to fast rhythms and mechanical limitations of the robot hands. Improving the RL method and hardware design could help address this. Secondly, the evaluation metric, F1 score, may not adequately capture musical performance and the position-based controller missing the target velocity would hinder the performance. Thirdly, our dataset includes only proprioceptive observations, whereas humans play piano using multi-modal inputs like vision, touch, and hearing; incorporating these could enhance the agent's capabilities. Furthermore, there are several challenges to deploying the learned agent on a real-world robot. This includes the challenges of obtaining the state of the piano and the hands (e.g., tracking the precise fingertip positions), optimizing a precise position controller at high speed as well as the sim-to-real gap for the highly dynamic piano-playing task, etc. Lastly, although we demonstrate better zero-shot generalization performance than RoboPianist-MT [4], there is still a gap between our best multi-task agent and RL specialists, which requires future investigation.

**Conclusion** In this paper, we propose a large-scale motion dataset named RP1M for piano playing with bimanual dexterous robot hands. RP1M includes 1 million expert trajectories for playing 2k musical pieces. To collect such a diverse dataset for piano playing, we lift the need for human-annotated fingering in the previous method by introducing a novel automatic fingering annotation approach based on optimal transport. On single songs, our method matches the baselines with human-annotated fingering and can be adopted across different embodiments. Furthermore, we benchmark various imitation learning approaches for multi-song playing. We report promising results in motion synthesis for novel music pieces when increasing the data size and identify the gap to achieving human-level piano-playing ability. We believe the RP1M dataset, with its scale and quality, forms a solid step towards empowering robots with human-level dexterity.

**Acknowledgments**

We thank the support of the Max Planck Institute for Intelligent Systems, Tübingen (Germany). We acknowledge CSC – IT Center for Science, Finland, for awarding this project access to the LUMI supercomputer, owned by the EuroHPC Joint Undertaking, hosted by CSC (Finland) and the LUMI consortium through CSC. Yi Zhao, Juho Kannala, and Joni Pajarinen acknowledge funding by the Research Council of Finland (345521 353138, 327911). We thank Yuxin Hou and Wenyan Yang for the insightful discussion. We thank all reviewers for their detailed and constructive comments.

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

# Appendix

## A  More Related Work

**Dexterous Robot Hands**  The research of dexterous robot hands aims to replicate the dexterity of human hands with robots. Many previous works [48, 49, 50, 51, 52, 53, 54, 55] use planning to compute a trajectory followed by a controller, thus require an accurate model of the robot hand. Closed-loop approaches have been developed by incorporating sensor feedback [56]. These methods also require an accurate model of the robot hand, which can be difficult to obtain in practice, especially considering the large number of active contacts between the hand and objects.

Due to the difficulty of actually modeling the dynamics of the dexterous robot hand, recent methods resort to learning-based approaches, especially RL, which has achieved huge success in both robotics [57, 58, 1] and computer graphics [59]. To ease the training of dexterous robot hands with a large number of degrees of freedom (DoFs), demonstrations are commonly used [60, 61, 62, 63, 64]. Due to the advance of both RL algorithms and simulation, recent work shows impressive results on dexterous hand manipulation tasks without human demonstrations. Furthermore, the policy trained in the simulator can further be deployed on real dexterous robot hands via sim-to-real transfer [65, 66, 67, 30, 68, 69, 70].

**Generalist Agents**  RL methods usually perform well on single tasks, however, as human beings, we can perform multiple tasks. Generalist agents are proposed to master a diverse set of tasks with a single agent [71, 57, 72, 73]. These methods typically resort to scalable models and large datasets [72, 74, 75, 76, 77]. Recently, diffusion models have achieved many state-of-the-art results across image, video, and 3D content generation [78, 79, 80, 81, 82] In the context of robotics, diffusion models have been used as policy networks for imitation learning in both manipulation [5, 75, 83, 84] and locomotion tasks [85]. The same technique has also been investigated in multi-task learning [75, 84]. We investigate the application of diffusion policy in high-dimensional control tasks, that is, playing piano with bimanual dexterous robot hands.

## B  RP1M Dataset Collection Details

### B.1  Reward Formulation

In Eq. (3) , we give the overall reward function used in our paper. We now give details of each term. $r_t^{\text{Press}}$ indicates whether the active keys are correctly pressed and inactive keys are not pressed. We use the same implementation as [4], given as: $r_t^{\text{Press}} = 0.5 \cdot (\frac{1}{K} \sum_t^K g(||k_s^i - 1||_2)) + 0.5 \cdot (1 - \mathbf{1}_{\text{fp}})$. K is the number of active keys, $k_t^i$ is the normalized key states with range [0, 1], where 0 means the $i$-th key is not pressed and 1 means the key is pressed. $g$ is tolerance from Tassa et al. [39], which is similar to the one used in Equation (2). $\mathbf{1}_{\text{fp}}$ indicates whether the inactive keys are pressed, which encourages the agent to avoid pressing keys that should not be pressed. $r_t^{\text{Sustain}}$ encourages the agent to press the pseudo sustain pedal at the right time, given as $r_t^{\text{Sustain}} = g(s_t - s_t^{\text{target}})$. $s_t$ and $s_t^{\text{target}}$ are the state of current and target sustain pedal respectively. $r_t^{\text{Collision}}$ penalizes the agent from collision, defined as $r_t^{\text{Collision}} = 1 - \mathbf{1}_{\text{collision}}$, where $\mathbf{1}_{\text{collision}}$ is 1 if collision happens and 0 otherwise. $r_t^{\text{Energy}}$ prioritizes energy-saving behavior. It is defined as $r_t^{\text{Energy}} = |\tau_{\text{joints}}|^\intercal |\mathbf{v}_{\text{joints}}|$. $\tau_{\text{joints}}$ and $\mathbf{v}_{\text{joints}}$ are joint torques and joint velocities respectively.

### B.2  Training Details

**Observation Space**  Our 1144-dimensional observation space includes the proprioceptive state of dexterous robot hands and the piano as well as L-step goal states obtained from the MIDI file. In our case, we include the current goal and 10-step future goals in the observation space (L=11). At each time step, an 89-dimensional binary vector is used to represent the goal, where 88 dimensions are for key states and the last dimension is for the sustain pedal. The dimension of each component in the observation space is given in Table 3.

Table 3: Observation space.

| Observations | Dim |
|---|---|
| Piano goal state | L · 88 |
| Sustain goal state | L · 1 |
| Piano key joints | 88 |
| Piano sustain state | 1 |
| Fingertip position | 3 · 10 |
| Hand state | 46 |

**Training Algorithm & Hyperparameters**   Although our proposed method is compatible with any reinforcement learning method, we choose the DroQ [40] as Zakka et al. [4] for fair comparison. DroQ is a model-free RL method, which uses Dropout and Layer normalization in the Q function to improve sample efficiency. We list the main hyperparameters used in our RL training in Table 4.

Table 4: Hyperparameters used in our RL agent.

| Hyperparameter | Value |
|---|---|
| Training steps | 8M |
| Episode length | 550 |
| Action repeat | 1 |
| Warm-up steps | 5k |
| Buffer size | 1M |
| Batch size | 256 |
| Update interval | 2 |
| Piano environment | |
| Lookahead steps | 10 |
| Gravity compensation | True |
| Control timestep | 0.05 |
| Stretch factor | 1.25 |
| Trim slience | True |
| Agent | |
| MLPs | [256, 256, 256] |
| Num. Q | 2 |
| Activation | GeLU |
| Dropout Rate | 0.01 |
| EMA momentum | 0.05 |
| Discount factor | 0.88 |
| Learnable temperature | True |
| Optimization | |
| Optimizer | Adam |
| Learning rate | 3e-4 |
| $\beta_1$ | 0.9 |
| $\beta_2$ | 0.999 |
| eps | 1e-8 |

### B.3 Computational Resources

We train our RL agents on the LUMI cluster equipped with AMD MI250X GPUs, 64 cores AMD EPYC "Trento" CPUs, and 64 GBs DDR4 memory. Each agent takes 21 hours to train. The overall data collection cost is roughly 21 hours * 2089 agents = 43,869 GPU hours.

### B.4 MuJoCo XLA Implementation

To speed up training, we re-implement the RoboPianist environment with MuJoCo XLA (MJX), which supports simulation in parallel with GPUs. MJX has a slow performance with complex scenes with many contacts. To improve the simulation performance, we made the following modifications:

- We disable most of the contacts but only keep the contacts between fingers and piano keys as well as the contact between forearms.
- Primitive contact types are used whenever possible.
- The dimensionality of the contact space is set to 3.
- The maximal contact points are set to 20.
- We use Newton solver with iterations=2 and ls_iterations=6.

After the above modifications, with 1024 parallel environments, the total steps per second is 159,376.

We use PPO implementation implemented with Jax to fully utilize the paralleled simulation. The PPO with MJX implementation is much faster than the DroQ implementation, which only takes 2 hours and 7 minutes for 40M environment steps on the Twinkle Twinkle Little Star song while as a comparison, DroQ needs roughly 21 hours for 8M environment steps. However, the PPO implementation fails to achieve a comparable F1 score as the DroQ implementation as shown in Fig. 5. Therefore, we use the DroQ implement with the CPU version of the RoboPianist environment.

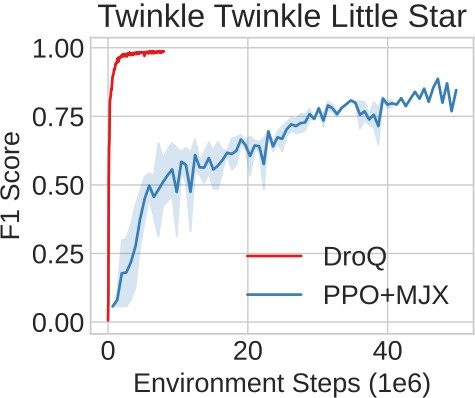

Figure 5: Comparison of the RL performance between DroQ and PPO with the MJX implementation of the RoboPianist environment. PPO+MJX is faster to run but has a worse performance than DroQ. We use DroQ with the CPU-version RoboPianist environment when training our RL agents.

## C   Multitask Benchmarking Details

A single multi-task policy capable of playing various songs is highly desirable. However, playing different music pieces on the piano results in diverse behaviors, creating a complex action distribution, particularly for dexterous robot hands with a large number of degrees of freedom (DoFs). This section introduces the baseline methods we have compared and the hyperparameters we have used. We also talk about the details of our multitask training and evaluation.

### C.1 Baselines and Hyperparameters

#### C.1.1 Behavior Cloning

Behavior Cloning (BC) [43] directly learns a policy using supervised learning on observation-action pairs from expert demonstrations, one of the simplest methods to acquire robotic skills. Due to its straightforward approach and proven efficacy, BC is popular across multiple fields. The method employs a Multi-Layer Perceptron (MLP) as the policy network. Given expert trajectories, the policy network learns to replicate expert behavior by minimizing the Mean Squared Error (MSE) between predicted and actual expert actions. Despite its advantages, BC tends to perform poorly in generalizing to unseen states from the expert demonstrations. The MLP we used features three hidden layers, each with 512 units, followed by Layer Normalization and an Exponential Linear Unit (ELU) activation function to stabilize training and introduce non-linearity.

Table 5: Hypermeters used in BC

| Hyperparameter | Value |
| --- | --- |
| Batch Size | 1024 |
| Optimizer | Adam |
| Learning Rate | 1e-4 |
| Observation Horizon | 1 |
| Prediction Horizon | 1 |
| Action Horizon | 1 |

#### C.1.2 BeT

Behavior Transformers (BeT) [44] uses a transformer-decoder based backbone with a discrete action mode predictor coupled with a continuous action offset corrector to model continuous actions sequences. It clusters continuous actions into discrete bins using k-means to model high-dimensional, continuous multi-modal action distributions as categorical distributions without learning complicated generative models. We adopted the implementation and hyperparameters from the Diffusion Policy codebase [5].

Table 6: Hyperparamerters used in BeT

| Hyperparameter | Value |
| --- | --- |
| Batch Size | 512 |
| Optimizer | AdamW |
| Learning Rate | 1e-4 |
| Num of bins | 64 |
| MinGPT n_layer | 8 |
| MinGPT n_head | 8 |
| MinGPT n_embd | 120 |
| Observation Horizon | 1 |
| Prediction Horizon | 1 |
| Action Horizon | 1 |

#### C.1.3 Diffusion Policy

Diffusion models have achieved many state-of-the-art results across image, video, and 3D content generation [78, 79, 80, 81, 82]. In the context of robotics, diffusion models have been used as policy networks for imitation learning in both manipulation [5, 75, 83, 84] and locomotion tasks [85], showing remarkable performance across various robotic tasks. Diffusion Policy [5] proposed to learn

an imitation learning policy with a conditional diffusion model. It models the action distribution by inverting a process that gradually adds noise to a sampled action sequence, conditioning on a state and a sampled noise vector. We evaluated both the U-Net-based Diffusion Policy (DP-U) and the transformer-based Diffusion Policy (DP-T). We build our diffusion policy training pipeline based on the original Diffusion Policy [5] codebase, which provides high-quality implementations.

Table 7: Hyperparameters used in DP-U

| Hyperparameter | Value |
|---|---|
| Batch Size | 1024 |
| Optimizer | AdamW |
| Learning Rate | 1e-4 |
| Weight Decay | 1e-6 |
| Diffusion Method | DDPM |
| Number of Diffusion Iterations | 100 |
| EMA Power | 0.75 |
| U-Net Hidden Layer Sizes | [256, 512, 1024] |
| Diffusion Step Embedding Dim. | 256 |
| Observation Horizon | 1 |
| Prediction Horizon | 4 |
| Action Horizon | 4 |

Table 8: Hyperparameters used in DP-T

| Hyperparameter | Value |
|---|---|
| Batch Size | 1024 |
| Optimizer | AdamW |
| Learning Rate | 1e-3 |
| Weight Decay | 1e-4 |
| Diffusion Method | DDPM |
| EMA Power | 0.75 |
| n_layer | 8 |
| n_head | 4 |
| n_emb | 156 |
| p_drop_emb | 0.0 |
| p_drop_attn | 0.3 |
| Observation Horizon | 1 |
| Prediction Horizon | 4 |
| Action Horizon | 4 |

## C.2  Training and Evaluation

We train the policies with 5 different sizes of expert data: 12, 25, 50, 100, and 150 songs, respectively. Subsequently, we assess the trained policies using two distinct categories of musical pieces. The first category, in-distribution songs, includes pieces that are part of the training datasets. Evaluating with in-distribution songs tests the multitasking abilities of the policies and checks if a policy can accurately recall the songs on which it was trained. The second group of songs for evaluation are out-of-distribution songs: those music pieces do not overlap with the training songs. The selected songs contain diverse motions and long horizons, making them challenging to play. This out-of-distribution evaluation measures the zero-shot generalization capabilities of the policies. Analogous

to an experienced human pianist who can play new pieces at first sight, we aim to determine if it is feasible to develop a generalist agent capable of playing the piano under various conditions.

Additionally, our framework is designed with flexibility in mind, allowing users to select songs not included in our dataset for either training data collection or evaluation. Furthermore, users have the option to assess their policies on specific segments of a song rather than the entire piece.

Table 9: In-distribution songs

| |
| --- |
| RoboPianist-etude-12-FrenchSuiteNo1Allemande-v0 |
| RoboPianist-etude-12-FrenchSuiteNo5Sarabande-v0 |
| RoboPianist-etude-12-PianoSonataD8451StMov-v0 |
| RoboPianist-etude-12-PartitaNo26-v0 |
| RoboPianist-etude-12-WaltzOp64No1-v0 |
| RoboPianist-etude-12-BagatelleOp3No4-v0 |
| RoboPianist-etude-12-KreislerianaOp16No8-v0 |
| RoboPianist-etude-12-FrenchSuiteNo5Gavotte-v0 |
| RoboPianist-etude-12-PianoSonataNo232NdMov-v0 |
| RoboPianist-etude-12-GolliwoggsCakewalk-v0 |
| RoboPianist-etude-12-PianoSonataNo21StMov-v0 |
| RoboPianist-etude-12-PianoSonataK279InCMajor1StMov-v0 |

Table 10: Out-of-distribution songs

| |
| --- |
| GiantMIDI-IsmagilovTimurSpringSketches-v0 |
| GiantMIDI-JohnsonCharlesLeslieGoldenSpiderRag-v0 |
| GiantMIDI-JoseffyRafaelValseDesDames-v0 |
| GiantMIDI-KiefertCarlPastoral-v0 |
| GiantMIDI-KleberHenryTheFancyPolka-v0 |
| GiantMIDI-KockKarlMelancholie-v0 |
| GiantMIDI-LackTheodoreMenuetDuXviiimeSiecleOp36-v0 |
| GiantMIDI-LecocqCharlesLeJourEtLaNuit-v0 |
| GiantMIDI-LefebureWelyLouisJamesAlfredApresLaChass-v0 |
| GiantMIDI-LindgreenCharlesFirstCourtship-v0 |
| GiantMIDI-LisztFranzRomanceOublieeS527-v0 |
| GiantMIDI-MacchiClaudioRomanzaSenzaParole1-v0 |
| GiantMIDI-MarcouPaulLeCosaqueOp15-v0 |
| GiantMIDI-MasonWilliamAPastoralNovelette-v0 |
| GiantMIDI-MattheyUlissePensieroOstinato-v0 |
| GiantMIDI-MayerlBillyJosephEgyptianSuite-v0 |
| GiantMIDI-MendelssohnFelixScherzoWoo2-v0 |
| GiantMIDI-MozartWolfgangAmadeusAllegroInCMajorK484-v0 |
| GiantMIDI-NegriCesareAlemanaDamore-v0 |
| GiantMIDI-OkellyJosephEn1795Op52-v0 |

