# OpenReview forum: "RP1M: A Large-Scale Motion Dataset for Piano Playing with Bi-Manual Dexterous Robot Hands"
_robot-learning.org/CoRL/2024/Conference — CoRL 2024_

### Official Review · Reviewer_Kzd5 · 2024-07-21
**Review of paper RP1M**

**Originality:** 3
**Technical Quality:** 4
**Clarity Of Presentation:** 4
**Potential Impact:** 3
**Recommendation:** 3
**Confidence:** 5

**Review:**

## Strengths

- Clarity of presentation
- Great related works section
- Pretty visualizations
- Open-source release of a massive dataset by robotics standards

## Weaknesses

- While I appreciate the simplicity and effectiveness of using optimal transport to solve for fingering information, and furthermore I understand how time-intensive the experiments can be, I think the experiments section can be improved overall. Specifically, I might even argue that you can remove 1 or 2 BC baselines (e.g. BC-RNN and IBC provide marginal value to the experiments) and focus more on improving the performance of Diffusion Policy and analyzing the failure modes. You have the gift of a huge dataset (1M snippets!) so if anything you should be really asking yourselves why performance is so paltry when the size of the dataset increases. There is either a data issue, or an optimization issue.
- The test set in the BC results seems pretty small? 12 for in-distribution songs, 5 for easy out-of-distribution, and 5 for hard out-of-distribution. I think you can increase it to get more statistical significance.
- Speaking of statistical significance, why are there no standard deviations in Table 2? Diffusion policy is non-deterministic right?
- Performance of BC methods decreases for H=4096 as the number of songs increases seems fishy. As I said above, this is fishy and leads me to believe there is some issue with the optimization or the data. Can you comment on this?
- Speaking of dataset, what if song snippets repeat? E.g. you chunked up songs into 20 second snippets. But chunks in the same song could have a lot of overlap (especially true in music where things are periodic). Did you account for that during the filtering process?

## Minor

* I appreciated the implementation in MJX and the explanation for falling back to CPU MuJoCo in the appendix.
* There a few grammar and spelling mistakes in the manuscript, please fix them.
* Figures and tables should be self-contained, please add a caption to each one of them, not just a title.

**Quality Of The Limitations Section:**

3

**Questions For Rebuttal:**

- It would be good to get some more analysis about the distribution of failures exhibited by the 2k specialist policies. Some questions that popped to mind:
  - Are there any instances where the actuator gains or energy penalty were limiting the policy from playing fast-paced songs?
  - Relatedly, how did you decide what DoFs to remove from the wrist? And if anything, why not add more DoFs like roll/pitch/yaw to the forearm?
  - You mention failures from mechanical limitations of the shadow hand. Why not filter out those MIDI files from the dataset to disentangle hardware from policy learning?
- I noticed you used a stretch factor 1.25 (mentioned in the appendix). Did you slow down the songs for a specific reason, maybe because it improved overall performance? If so, is this a fair comparison with the RoboPianist curves?
- What software suite did you use for the BC experiments? Were all the baselines implemented using robomimic/diffusion policy or just diffusion policy used Chi et al's codebase?
- It would be nice to get a bit more details regarding how you formulated the optimal transport problem. Was any tweaking done to get it to work? I can imagine one could speedup the solver by ignoring keys on the far right/far left of the piano, or basically only considering keys that are within a horizon of time into the future? Also did you consider another fingering algorithm baseline, like the ones mentioned in the related works section?

**Robotics Focus:**

3

**Summary Of Paper:**

This paper addresses a big limitation of the recently introduced RoboPianist benchmark: the need for fingering labels to train a piano playing policy with RL. They solve for fingering labels by formulating it as an optimal transport problem. Using this technique, they label 2000 MIDI files and demonstrate that they can effectively train specialist policies per MIDI file to generate a dataset of 1M piano playing motions. Finally, they distill this data into a multitask policy using diffusion policy and compare with other BC baselines.

**Summary Of Recommendation:**

Overall a good paper that is well written and provides an elegant solution to the piano fingering problem. As it stands, I recommend accepting this paper, but I have some concerns regarding the BC results / dataset. I look forward to the authors clarifying my concerns.

---

### Official Review · Reviewer_Sw5X · 2024-07-27
**Review for "RP1M: A Large-Scale Motion Dataset for Piano Playing with Bi-Manual Dexterous Robot Hands"**

**Originality:** 3
**Technical Quality:** 3
**Clarity Of Presentation:** 3
**Potential Impact:** 3
**Recommendation:** 3
**Confidence:** 3

**Review:**

The paper focuses on a dynamic task with dexterous bimanual robot hands that requires precise contact-rich control. This task comes across as an important step in developing complex manipulation capabilities in robotics and is a step in the right direction. The background and related works section in the paper seems to be exhaustive and well written.
For benchmarking of the results on the provided dataset, the paper uses a number of baselines that include variants of behavior cloning and diffusion policy.

As shown in the results, diffusion policy beats the performance of other baselines by a significant margin. Although the experiments have been done with a variety of settings, the discussion sections lack the explanation behind the obtained results. The experiments section could also improve from the inclusion of more exhaustive set of baselines that could include baselines that are based on transformers (for e.g., behavior transformer [1], PACT [2]), LfD (Robomimic [3]), IL+RL (Relay Policy Learning [4]). This would provide the readers about how these baselines perform on the proposed benchmark.

Also, it would be useful for the reader if the authors could compare with methods that utilize a small number of human demonstrations. Even using videos available on the internet of people playing a piano could be used as a learning signal for the model.
The analysis of the dataset statistics along with the quantitative results showing the performance improvement over method without fingering is thorough and convincing.

While this work is an important step in the direction of human-level dexterity, it would be more convincing if the authors could justify that by displaying some results on a real-world robot too. The transfer of the robot policy learnt in simulation to a real-world scenario would justify the utility of the proposed dataset to enable dexterity in robots.
Finally, the dataset proposed, which is the a large scale dataset for dynamic bimanual manipulation with dexterous hands could serve the robotics community in a positive way, but the paper could certainly improve with more elaborate evaluation.

[1] Behavior Transformers: Cloning k modes with one stone. Nur Muhammad Mahi Shafiullah, Zichen Jeff Cui, Ariuntuya Altanzaya, Lerrel Pinto

[2] PACT: Perception-Action Causal Transformer for Autoregressive Robotics Pre-Training. Rogerio Bonatti, Sai Vemprala, Shuang Ma, Felipe Frujeri, Shuhang Chen, Ashish Kapoor

[3] What Matters in Learning from Offline Human Demonstrations for Robot Manipulation. Ajay Mandlekar, Danfei Xu, Josiah Wong, Soroush Nasiriany, Chen Wang, Rohun Kulkarni, Li Fei-Fei, Silvio Savarese, Yuke Zhu, Roberto Martín-Martín

[4] Relay Policy Learning: Solving Long-Horizon Tasks via Imitation and Reinforcement Learning. Abhishek Gupta, Vikash Kumar, Corey Lynch, Sergey Levine, Karol Hausman

**Quality Of The Limitations Section:**

3

**Questions For Rebuttal:**

I would want the authors to consider adding more exhaustive baselines to the evaluation for the reader to get an idea about how the diverse set of approaches available work on the proposed benchmark. It would also be useful to see some real-world transfer experiments on this challenging task of robot piano playing.

**Robotics Focus:**

3

**Summary Of Paper:**

The paper uses dexterous robot hands and introduces a new dataset, called the Robot Piano 1 Million (RP1M) dataset, containing over a million bi-manual piano playing trajectories. This dataset aims to improve performance of robots in the task of piano playing using imitation learning, facilitated by framing finger placements as an optimal transport problem.  The main contributions of the paper include the introduction of a novel dataset for an interesting, complex and previously under-researched task of piano playing, formulating fingering as an optimal transport problem that enables the authors to collect diverse large scale data using RL.

**Summary Of Recommendation:**

I feel the paper is a step in the right direction towards improving the robotics manipulation with dexterous arms and provides an important benchmark for the robotics community. The lack of exhaustive testing of prior approaches and real-world evaluation could be circumvented by the authors and could improve the overall quality of the paper.

---

### Official Review · Reviewer_EzFs · 2024-07-28

**Originality:** 3
**Technical Quality:** 3
**Clarity Of Presentation:** 3
**Potential Impact:** 2
**Recommendation:** 3
**Confidence:** 4

**Review:**

Strength:
 - The paper addresses an interesting problem: enabling robots to play piano songs. This task demands precise hand-finger movements and involves significant dynamism.
 - The use of optimal transport-based rewards for robotic piano playing is, to the reviewer’s knowledge, a novel approach.

Weakness:
 - The main contribution of the paper is the introduction of a domain-specific dataset for robotic piano playing. While the task itself is interesting, the domain remains quite specialized, making it challenging to assess how the dataset could impact other robotics tasks/domains.
 - The quality of the RL-generated dataset can be highly variable, and a more comprehensive evaluation mechanism is needed. Currently, the dataset is collected by running an agent after training for a large number of steps. It would be beneficial to implement a data filtering mechanism to ensure the quality of the collected dataset or to only collect data after the RL agent reaches a certain accumulated reward threshold. For instance, section 4 mentions that “agents with an F1 score ≥ 0.75 are capable of playing sheet music reasonably well with only minor errors.” In this case, filtering out agents or retraining those with an F1 score less than 0.75 is important to maintain dataset quality.

**Quality Of The Limitations Section:**

3

**Questions For Rebuttal:**

- How to measure the generated data quality and what can be a good data filtering mechanism?

**Robotics Focus:**

3

**Summary Of Paper:**

The paper introduces the Robot Piano 1 Million (RP1M) dataset, which contains bi-manual robot piano-playing motion data for 2k sounds. Considering that the human annotation required in the RoboPianist is hard to collect, it proposes an Optimal-transport-based reward when training the RL specialist without human annotated fingering. To collect the motion dataset, it first trains RL agents for each song and then uses the trained RL agents to collect datasets. It further benchmarks Imitation Learning methods based on the collected dataset.

**Summary Of Recommendation:**

The paper focuses on a specific task domain and generates a valuable dataset for that domain.

---

### Official Review · Reviewer_3sJW · 2024-07-28
**Clear and well supported work which would benefit from ablations**

**Originality:** 3
**Technical Quality:** 4
**Clarity Of Presentation:** 5
**Potential Impact:** 3
**Recommendation:** 3
**Confidence:** 4

**Review:**

This work presents a well-curated dataset for training dextrous, bi-manual piano-playing policies. The overall paper is clear and the simulation experiments are complete but could be expanded with more ablations. There is no demonstration of the method on hardware which would greatly improve the paper, but I do not think is required.

Strengths:

-Overall, this paper is very clear and clearly articulates the method and intuition with a few exceptions

-The method does not rely on human annotations and is hand morphology agnostic allowing this method to scale better than alternative methods

-The analysis of the learned fingering due to embodiment constraints is very interesting and could benefit from further detail

Weaknesses:

-It would be helpful to further elaborate on in section 4.1 the design of reward equation 2

-The work currently uses DroQ but could benefit from an ablation study with different RL algorithms

-The work should also include ablations for hyperparameters of DroQ to ensure baselines have the best possible performance they can attain

-For Section 5, it would be helpful to further discuss why diffusion policies don’t perform as well as the generalist agent instead of keeping that for future work as it is very interesting to see

-I would recommend expanding on Figure 4 by adding more statistics about the dataset and removing Figure 3 if necessary

-The limitations are good but I would add some details about expected problems when deploying on real hands

Consider to cite the following additional work:
[1] also trains a general piano-playing agent from demonstrations, but they curate their dataset from YouTube videos and have a framework to extract features from the videos, train expert policies for each song, and distill the policies into one agent. Since this work can be used with different hand embodiments, [2] is relevant as it trains a diffusion policy with biological constraints from an anthropomorphic hand and is competitive with SOTA RL results. The authors claim that this dataset is the “first large-scale dataset of dynamic, bi-manual manipulation with dexterous robot hands”, however, [3] also creates “a dataset of two hands that dexterously manipulate objects” albeit from video frames. This claim should be reviewed and ensured it is accurate.

[1] https://arxiv.org/abs/2407.18178

[2] https://spj.science.org/doi/full/10.34133/cbsystems.0104

[3] https://arxiv.org/abs/2204.13662

**Quality Of The Limitations Section:**

2

**Questions For Rebuttal:**

-Why can [3] not be considered the “first large-scale dataset of dynamic, bi-manual manipulation with dexterous robot hands”? It seems like this is a strong claim and I would welcome more elaboration for this as it is a strong statement.

-In the related works, it would be helpful to further distinguish from [1] as they also train an RL policy for each song but then distill the policies to create a generalist policy rather than creating a dataset to train a single policy.

-What additional metrics can be included to characterize the dataset?

**Robotics Focus:**

3

**Summary Of Paper:**

This work proposes a bi-manual, dexterous, piano-playing dataset with 1M demonstrations. The dataset is diverse by forgoing human annotations and utilizing an optimal transport reward for training the RL agent. The work is competitive with human-annotated methods for individual songs and provides preliminary results for a multi-task piano-playing policy.

**Summary Of Recommendation:**

Overall, this dataset will be helpful to the community and the paper is both clear and has sufficient experimental support. However, the paper would benefit greatly from additional ablations and deployment on real world hands as that would clearly show the usefulness of this simulation dataset.

---

### Official Review · Reviewer_qnEm · 2024-07-28

**Originality:** 3
**Technical Quality:** 3
**Clarity Of Presentation:** 3
**Potential Impact:** 3
**Recommendation:** 3
**Confidence:** 4

**Review:**

Strengths:
1. This is the first large-scale dexterous robot piano playing dataset, generated with RL and does not require human annotations. It is a good resource for this line of research.
2. It is refreshing to see the proposed method is compatible with different embodiments with either four or five fingers.
3. The proposed method is able to handle highly dynamic pieces like "Flight of the Bumblebee," which is truly amazing.
4. The benchmark of multi-task imitation learning based on RP1M highlights the strength of diffusion policy in such a multi-modal case, which is a good finding for multi-task dexterous manipulation.


Questions:
1. My main concern lies in the simplicity of the position-based controller. For piano playing, it seems critical to also align the velocity of the finger movements, not just the positional mapping (Line 150), since it will affect the strength of the sound and should be an important metric of music playing. This becomes more important if we plan to use this dataset on a real-world robot with PD controllers.
2. For multi-task imitation learning, MT-ACT from RoboAgent[1] is an important missing citation and baseline method in the benchmark. It would be very helpful to show MT-ACT’s performance in the dexterous manipulation domain.
3. As a dataset paper, I would recommend the authors use a website for browsing the quality of the data. The current video submission only contains a few pieces, making it hard to estimate the scale and quality of the dataset.
4. Since RP1M contains so many pieces of piano playing, is it possible to incorporate generative models for music composition and synthesizing? If not, what might be the remaining challenges? More data or better model architecture?
5. It would also be beneficial to explain how to use such a dataset on a real-world robot system. For example, how to track the precise fingertip positions? Is there a sim-to-real gap during highly dynamic pieces? Is positional control enough for piano playing in the real world? I’m not sure whether this year's CoRL has the rule of real-world robot testing results. But I do agree that reproducing the system in the real world requires too much engineering effort.

[1]. RoboAgent: Towards Sample Efficient Robot Manipulation with Semantic Augmentations and Action Chunking

**Quality Of The Limitations Section:**

3

**Questions For Rebuttal:**

Same as the prior question section

**Robotics Focus:**

3

**Summary Of Paper:**

This work introduces the RP1M dataset, which provides extensive bi-manual piano playing data with over one million trajectories. By matching finger positions, the dataset enables the automatic annotation of songs, significantly reducing the need for human-labeled data and making it feasible to scale imitation learning.

**Summary Of Recommendation:**

Overall, this work is a good contribution to the community. However, due to the missing baseline and the potential challenge of the position-based controller, I will start with a weak reject and will adjust my rating based on the authors' response.

---

### Official Review · Reviewer_cZsS · 2024-07-29
**Good dataset on a fairly unexplored topic**

**Originality:** 4
**Technical Quality:** 3
**Clarity Of Presentation:** 4
**Potential Impact:** 3
**Recommendation:** 3
**Confidence:** 4

**Review:**

The paper presents a new dataset for piano playing, the dataset consists of 2k songs. The paper evaluates several imitation learning baselines and shows that having finger assignments is more beneficial than using ground truth annotations. Further, it shows that diffusion policy is superior to all other baselines but not very sample efficient.

Strengths

- Largest dataset available for this task with 2k songs.
- Evaluations on cross embodiment, reducing training data, benchmarking multiple imitation learning methods
- Made data collection more efficient by removing the need for finger information and replacing the module with optimal transport where we cast the assignment problem of assigning finger to each stroke


Weaknesses

- Evaluations can be improved -- for instance the number of methods on imitation learning
- Model improvements were not explored well. The numbers in table 2, show that the diffusion policy's performance decreases as we add more weights. The paper could have also benefitted from model-based methods
- The numbers produced in table 2 are a bit problematic, mainly because the diffusion model performance reduces with increase in data

**Quality Of The Limitations Section:**

3

**Questions For Rebuttal:**

Please confirm if the dataset would be released in public.

**Robotics Focus:**

3

**Summary Of Paper:**

The paper presents a new dataset for piano playing. The dataset consists of 2k songs. The paper makes data collection a bit cheaper by using Optimal Transport to assign finger to keys such as the overall finger movement is minimized.

**Summary Of Recommendation:**

I am recommending to accept the paper mainly because I like the dataset and it is an unexplored problem.

---

### Official Review · Reviewer_iGED · 2024-07-31
**Good job on building and expanding upon an existing benchmark, but the benchmark itself is problematic.**

**Originality:** 3
**Technical Quality:** 3
**Clarity Of Presentation:** 5
**Potential Impact:** 2
**Recommendation:** 3
**Confidence:** 4

**Review:**

## Post-rebuttal update

In the conversation below, the authors convinced me that this paper has enough contributions to warrant publication, despite shortcomings in the benchmark itself. I think I've made my point about applicability of this benchmark to real piano playing, but don't want to block the author's contribution to robotics and open source code.

I have updated my recommendation to weak accept.

## Original review

Clarity:

The paper is written clearly and reads smoothly. It was easy to understand what the authors did.

Originality:

Within the field of robotics, this paper seems unusual and original. Given the goal of optimizing the "F1 score" (from (Zakka et. al., 2023)), reformulating fingering as an optimal transport is clever, and the fact that RL is still used allows agents to use the fingering as a hint, so imperfect or implausible finger annotations are not a complete blocker for performance.

This work contributes to a body of work around automatic piano fingering generation, which I'm not qualified to comment on. I think the originality of the proposed approach should be evaluated compared to that body of work.

Significance:

Again, the work can be evaluated as a robotics paper, or in its merics for automatic piano fingering generation.
In terms of robotics, the method is quite specific to piano playing, and to optimizing the particular metric introduced in Zakka's paper. It will be of interest to others in the field who want to try their hand at the benchmark, but perhaps not to the wider community.
For automatic piano fingering generation, it's possible that with small modifications this work could be a competitive approach for generating piano fingerings: use a realistic human hand model and RL training to improve upon the imperfect fingering generated by the optimal transport heuristic.

Strengths:

- The work greatly expands on the existing piano playing benchmark and presents an approach that is applicable to any piece of piano music.
- They demonstrate an ability of learned agents to generalise beyond individual pieces that they were trained on.

Weaknesses:

- The work inherits the main weakness of the original benchmark, which is nothing to do with robotics, and all to do with music. The F1 metric is not grounded in musical insight, and a near-perfect F1 score does not correspond to a good musical performance. Specifically, the perceived quality of a piano performance depends on the precise timing of the start of each note, and on the velocity with which they are played, neither of which are taken into account by the metric. Additionally, controlling the robot and evaluating the rewards with a 50ms timestep, regardless of the underlying music's rhythm, guarantees that notes will be missed. For example, in Flight of the Bumblebee, each note's duration is about 80ms, so an error of +-25ms represents a significantly incorrect timing. In general, listeners will notice a latency of 15ms. As such, I don't have confidence that the recorded trajectories in this dataset correspond to useful musical performances. This can be heard in the demonstrated performances in the supplementary video. This is not the authors' fault, but limits the usefulness of the data for anything but this particular benchmark.

- When proposing a new way of generating piano fingerings, it seems important to compare it to other automatic fingering generation methods, such as those cited in the introduction. The paper currently only compares to having no fingering charts at all.

- It is not clear how closely the expert agents actually follow the generated fingering. Is its importance in guiding the agents towards a good initial guess, or is it really good enough for performing the pieces? I would like to see some measure of the difference between suggested fingerings and performed fingerings.

Minor comments:

- I think that "bimanual" is more common than "bi-manual". Similarly for "multimodal" vs "multi-modal".
- On line 70, you argue that RL is used because of the difficulty of modelling the dynamics of a hand accurately. This might be a reason for model-free RL in some domains, but in this benchmark there is no modelling error: one could plan and execute in the simulated environment with perfect fidelity. The reason to use RL for this work is because the reward landscape is very challenging for gradient-based trajectory optimization methods, with the large number of possible self-collisions that a bimanual setup has.
- In the final slide of the supplementary video, the audio and video are not well synchronized.
- In figure 4, "Num. of Active Keys Per Musical Piece" is a strange metric to have. Is this the number of notes played in the piece?

**Quality Of The Limitations Section:**

2

**Questions For Rebuttal:**

- Address the downsides of the benchmark in the limitations section.
- Compare the proposed fingering generation method to others in the literature.
- Check whether the generated fingering is followed by the RL specialists.
- Please address the minor comments listed above.

Request to area chair: Maybe reach out to reviewers outside of robotics, who wrote about piano fingering generation.

**Robotics Focus:**

2

**Summary Of Paper:**

The authors propose a method for automatically generating fingering charts for piano music, by posing the problem of mapping fingers to piano keys as an "optimal transport" problem that can be solved with an off-the-shelf solver. They show that giving these fingering charts to RL agents and adding an auxiliary reward for following the fingering sequence greatly improves the agents' ability to learn to play a song. The authors present a new dataset, RP1M, which is the result of training a specialist agent for each of 2k songs, and rolling out 500 trajectories from each agent. The authors demonstrate that this approach works for 4-finger hands, and that the data can be used to train piano-playing agents using various imitation-learning methods. The environment and evaluation metric are based on the benchmark presented by (Zakka et. al., 2023).

**Summary Of Recommendation:**

Because of the limitations of the benchmark itself, I don't think this work will have impact on real robot piano playing. That said, it is a big improvement on the existing benchmark, and we should accept it on the merits of its robotics work.

---

### Decision · Program_Chairs · 2024-09-04

**Decision:**

Accept

**Comment:**

# strengths
- The strength of the paper has been well recognized by all reviewers.
- Generalization ability of learned agents to beyond individual pieces has also been well recognized.
- The novelty of finger placements as an optimal transport problem has been highlighted
- Evaluations on cross-embodiment has been welcomed by multiple reviewers

# weakness
- There are concerns raised on the evaluation metrics and their relevance to realistic piano playing.
- Details missing on baselines and ablation. They additionally are in need of further clarifications

# recommendation
Reviewers have provided an overwhelming diversity of suggestions for improvements. Authors are encouraged to improve the submission based on the provided suggestions.

# rebuttal summary
The authors provided comprehensive responses to the reviewer's feedback. The authors also proposed to add clarifications, improve on baselines, and make the datasets public. Post these clarifications, reviews adjusted their scores. All reviewers are aligned on their recommendation for the paper. Based on their suggestions, AC recommends to accept the paper.